# A Short-Term Supplementation with a Polyphenol-Rich Extract from Radiata Pine Bark Improves Fatty Acid Profiles in Finishing Lambs

**DOI:** 10.3390/ani13020188

**Published:** 2023-01-04

**Authors:** Nelson Vera, Sandra Tatiana Suescun-Ospina, Rodrigo Allende, Constanza Gutiérrez-Gómez, Tania Junod, Pamela Williams, Cecilia Fuentealba, Jorge Ávila-Stagno

**Affiliations:** 1Departamento de Ciencia Animal, Facultad de Ciencias Veterinarias, Universidad de Concepción, Campus Chillán, Chillán 3812120, Chile; 2Escuela de Ciencias Animales, Universidad de los Llanos, Vereda Barcelona, Villavicencio 500017, Colombia; 3Escuela de Medicina Veterinaria, Facultad de Recursos Naturales y Medicina Veterinaria, Universidad Santo Tomás, Santiago 4030000, Chile; 4Departamento de Producción Animal, Facultad de Agronomía, Universidad de Concepción, Campus Chillán, Chillán 3812120, Chile; 5Unidad de Desarrollo Tecnológico, Universidad de Concepción, Coronel 4190000, Chile; 6Centro Nacional de Excelencia para la Industria de la Madera (CENAMAD), Pontificia Universidad Católica de Chile, Santiago 7820436, Chile

**Keywords:** tannins, animal performance, blood metabolites, carcass, meat, ruminants

## Abstract

**Simple Summary:**

A polyphenolic-rich extract from radiata pine bark (PBE) has been demonstrated to reduce in vitro ammonia nitrogen concentration without detrimentally affecting diet digestibility or ruminal fermentation parameters, suggesting that PBE can improve dietary protein use efficiency by increasing the protein outflow to the intestine. This in vivo study evaluates the effects of a short-term (21-d) supplementation with PBE at three concentrations (0, 1, and 2% dry matter basis) on animal performance, blood parameters, and fatty acid (FA) profiles in finishing Suffolk lambs. The supplementation with PBE improved sheep FA profile by (i) reducing the saturated FA and the Σn-6/Σn-3 ratio, (ii) increasing unsaturated FA, and (iii) increasing FAs that are beneficial to human health. Additionally, PBE supplementation enhances relative growth rate and dietary protein conversion without affecting intake, growth performance, blood parameters, or carcass traits.

**Abstract:**

The aim of this study was to evaluate the effects of a short-term supplementation with a polyphenol-rich extract from radiata pine bark (PBE) on animal performance, blood parameters, and fatty acid (FA) profiles in finishing lambs. Twenty-seven Suffolk lambs (4 months old) fed a finishing diet were randomly assigned to one of the following treatments: diet without PBE or diet supplemented with PBE on a 1 or 2% dry matter (DM) basis, for 35 d (14 d adaptation and 21 d of experimental period). Data were compared using Tukey’s test and orthogonal and polynomial contrasts. The results indicated that the supplementation with PBE increased (*p* = 0.008) relative growth rate (RGR) and improved (*p* = 0.003) protein conversion (CPC), whereas weight gain, carcass characteristic, and blood parameters were unaffected (*p* ≥ 0.106). Total mono- and polyunsaturated FAs, conjugated linoleic acid (CLA), and vaccenic and oleic acids were linearly increased (*p* ≤ 0.016) by PBE supplementation. In contrast, total saturated FAs (ΣSFA), Σn-6/Σn-3 ratio, atherogenicity index (AI), thrombogenic index (TI), and the proportion of elaidic acid were linearly decreased (*p* ≤ 0.018). In conclusion, the supplementation with 1 or 2% DM of PBE improves subcutaneous FA profiles by increasing CLA and reducing ΣSFA, Σn-6/Σn-3 ratio, AI, and TI. Additionally, PBE supplementation has the potential to improve RGR and CPC, with unaffected intake, growth performance, blood parameters, or carcass characteristics.

## 1. Introduction

Radiata pine (*Pinus radiata* D. Don) bark is a very abundant forestry by-product with negative environmental impacts when used as fuel (burned to provide heat and energy) or deposited directly on soil [1]. However, radiata pine bark (PB) is also a rich source of phenolic compounds, including catechin, epicatechin, quercetin, dihydroquercetin, taxifolin, phenolic acids, and condensed tannins (CT), which are flavonoid polymers formed from catechin and epicatechin, with varying degrees of condensation [2].

Previous studies have demonstrated that when ruminants are supplemented with plants, plant parts, or extracts rich in polyphenols, tannins, or CT in low-to-moderate concentrations, it results in numerous positive effects on livestock production, such as (i) increased live weight (LW), milk production, and wool growth; (ii) improved health (antibloat and anthelmintic properties) and nutritional (reduced protein degradation, and increased undegraded protein flow to the small intestine) status of ruminants [3]; (iii) reduced environmental impact by reducing methane emissions and urinary nitrogen (N) excretion [4]; and (iv) changes in fatty acid (FA) profile, enriching meat and milk with health-promoting FAs by modulating FA biohydrogenation (BH) and increasing rumenic acid (*c*9,*t*11-18:2; RA), the most abundant conjugated linoleic acid (CLA) isomer, as well as vaccenic acid (*t*11-18:1; VA) at the expense of saturated FAs (SFA; [5]). However, information on the use of CT derived from PB in in vivo studies is limited to a few trials with goats. As a ground ingredient (maximum 30% of diet), *P. taeda* bark resulted in improved performance, a favorable modulation of rumen fermentation [6], a positive impact on N-balance [7], and moderation of the poly- and monounsaturated FA (PUFA and MUFA, respectively) accumulation in goat meat [8]. More recently, substitution of 30% PB (*Pinus* spp.) for bermudagrass did not affect blood metabolites or carcass traits in meat goats [9], and when used as a supplement (1.2% of LW), *P. taeda* bark did not affect performance or meat quality [10].

A polyphenolic-rich extract from radiata PB [11] has consistently demonstrated that in low concentrations (≤2% of dry matter [DM]) it reduced concentrations of ammonia nitrogen (NH_3_-N) by 50% [12], 30.9% [13], and 35.9% [14], without detrimentally affecting diet digestibility or ruminal fermentation parameters, which suggests that radiata PB extract (PBE) supplementation can improve dietary protein use efficiency by increasing the protein outflow to the intestine. Tannins’ effects on rumen proteolysis have been attributed to their capability to form tannin–protein complexes [15] and reduce dietary protein breakdown. These complexes are dissociated at low pH in the abomasum [16] and small intestine, thus increasing amino acid availability for digestion and absorption in the lower gut [10] and improving animal performance [3]. Effects of radiata PBE on rumen microorganisms and enzymes, rumen lipolysis, and BH have not been assessed. As such, the potential improvement of FA profiles of ruminant-derived foods [5,8,17] should be tested.

We hypothesized that the incorporation of a 1 or 2% DM basis of radiata PBE in lamb diets could improve animal performance and change FA profiles without affecting DM intake (DMI) or animal health. The objective of this study was to evaluate the effects of a short-term supplementation with radiata PBE on the growth performance, feeding efficiency, blood parameters, carcass traits, and subcutaneous FA profiles in finishing Suffolk lambs.

## 2. Materials and Methods

### 2.1. Animal Care and Management

The study was conducted at the Livestock Systems and Nutrition Laboratory of the Universidad de Concepción (UdeC), Chillan, Chile. Care, management, and animal assistance were certified by the UdeC animal ethics and welfare committee (CBE 28–2019).

Lambs were stratified by LW and randomly assigned into separate pens (1.2 m^2^) in a vented barn, vaccinated (Clostribac-8^®^, Pfizer, Chile), and dewormed with ivermectin (Ivermic^®^, Microsules, Barros Blancos, Uruguay) and nitroxynil (Dovenix^®^, Merial, Chile). Clean fresh water was permanently available for all animals.

### 2.2. Experimental Procedure and Dietary Treatments

A total of 27 Suffolk lambs (17 males (M) and 10 females (F)), average 4 months old and 34.6 ± 4.4 kg LW, were randomly assigned to one of three dietary treatments to undergo a 35 d feeding trial. The experimental groups of 9 lambs each were formed as follows: control (*n* = 6 M and 3 F), 1 (*n* = 6 M and 3 F), and 2 (*n* = 5 M and 4 F).

The radiata PBE (13.3% polyphenols; 4.35% tannins) is an aqueous solution (38.0% DM) that was prepared by methanolic extraction at the Technological Development Unit—UdeC. The base diet was a mixture of alfalfa hay (*Medicago sativa*), soybean meal, wheat bran, corn grain, and mineral supplement (Table 1). All ingredients were ground (Grain Mill, Breuer, Temuco, Chile) to 5 cm particle size before adding radiata PBE at concentrations of 1 and 2% DM [14] and thereafter mixing. The three dietary treatments were control (diet without PBE), diet supplemented with PBE at 1% DM, and diet supplemented with PBE at 2% DM. All dietary treatments met requirements of lambs for weight gain and growth according to NRC [18]. This formulation was developed to evaluate the radiata PBE effect on dietary protein conversion.

### 2.3. Chemical Composition of Diets

Samples of each dietary treatment were collected weekly and were chemically analyzed at the Animal Nutrition Laboratory—UdeC. Dry matter (Method 934.01), total ash (Method 942.05), crude protein (CP; Method 954.01), and ether extract (EE; Method 920.39) were determined (in triplicate) according to the AOAC [20]. Neutral detergent fiber was determined according to Mertens [21] using heat-stable amylase solution, and expressed inclusive of residual ash (aNDF). Acid detergent fiber (ADF) was determined by method 973.18 of AOAC [20], whilst the total polyphenol (TP) and tannin (TT) contents were determined according to Makkar [22]. For radiata PBE, DM and ADF were determined by the aforementioned procedures, and lignin was determined according to AOAC [20] method 973.18. Organic matter (OM) was estimated by the difference between DM and total ash, whereas hemicellulose (HE) by the difference between the aNDF and ADF. The nonfibrous carbohydrates (NFC) were estimated according to Hall [19].

Lipids from the experimental diets were extracted according to Bligh and Dyer [23] and methylated according to Hartman and Lago [24]. Fatty acid methyl esters (FAME) were analyzed by gas chromatography (GC) in an Agilent 7890B (Agilent Technologies, Inc., Santa Clara, CA, USA), equipped with a flame ionization detector (FID) as described by Villeneuve et al. [25].

### 2.4. Determination of Dry Matter Intake and Performance

The lambs were fed daily ad libitum (08:00 a.m.). Diet offers and rejections were weighed daily to determine DMI, adjusting offers to maintain rejection rate at 10%. Lamb DMI was used to determine nutrient intake according to diet compositions. All lambs were weighed weekly before morning feeding, average daily gain (ADG, g/d), feed conversion ratio (FCR, g/g), and CP conversion ratio (CPC, g/g), according to Berry and Crowley [26]. Furthermore, the relative growth rate (RGR) for lambs was calculated according to Fitzhugh and Taylor [27], and considered as an indirect measure of efficiency, since it does not use the individual animal feed intake data [27].

### 2.5. Blood Collection and Analysis

At the end of the experimental period, individual blood samples (4 mL) were collected from the jugular vein in Vacutainer tubes (BD, Franklin Lakes, NJ, USA) with and without anticoagulant ethylenediaminetetraacetic acid (EDTA), and immediately sent to the Clinical Laboratory—UdeC to perform a hematological and serum biochemical profile, according to the techniques described by Moritz [28] and Smith [29], respectively.

### 2.6. Slaughter and Carcass Evaluation

After the blood collection, lambs were slaughtered at a commercial abattoir following a 16 h fasting period, in accordance with the protocols established by the Chilean regulation [30,31]. Hot carcass weight (HCW, kg) was recorded immediately after slaughter, and cold carcass weight was recorded 24 h after cooling at 4 °C. Hot carcass yield (HCY, %), cold carcass yield (CCY, %), and weight loss by cooling (WLC, %) calculations were performed according to Maciel et al. [32].

### 2.7. Lipid Extraction and Analysis

Subcutaneous adipose tissue covering the *Longissimus dorsi* muscle between 10th and 13th ribs were sampled from the cold carcasses 24 h after slaughter, and immediately individually vacuum-packed in polyethylene bags and frozen (−20 °C) until analysis. Fatty acids were extracted from subcutaneous fat, as described by Folch et al. [33]. Samples of 1 g were homogenized with 20 mL of GC-grade hexane and isopropanol (3:2 *v*/*v*) solution using Stuart handheld homogenizer (SHM1, UK) set at 10,000 rpm for 1 min, and a second homogenization was performed adding GC-grade hexane (10 mL) to the mixture. Then, the mixture was centrifuged (10 min, 3000 rpm), and lipids were methylated using a cold methanolic solution of potassium hydroxide 0.5 N HCl and nonadecanoic acid (19:0) methyl ester (3.3 mg/mL of hexane) as an internal standard [34]. Fatty acid methyl esters were analyzed by GC in an Agilent 7890B (Agilent Technologies, Inc., Santa Clara, CA, USA), equipped with an FID (detector temperature 250 °C), according to David et al. [35]. The initial oven temperature (120 °C) was held for 1 min, increased by 10 °C/min to 175 °C, held for 10 min, and increased at 5 °C/min to 210 °C, held by 5 min, increased at 5 °C/min to 230 °C and finally held for 5 min. Helium was used as the carrier gas (head pressure 32 psi and flow rate of 2 mL/min). The concentrations of FAME were expressed as a relative percentage of the total content of FA (as g/100 g of FA).

To assess the nutritional quality of the lipid tissue, the atherogenicity index (AI) and thrombogenic index (TI) were calculated according to Ulbricht and Southgate [36], whereas the hypocholesterolemic and hypercholesterolemic (h:H) ratio was determined according to Santos-Silva et al. [37], and the desirable FAs according to Rhee [38]. The activities of delta-9-desaturase C16 (Δ^9^C16), delta-9-desaturase C18 (Δ^9^C18), and elongase were estimated according to Smet et al. [39]. Total trans FAs (TFA) were calculated, excluding VA and total CLA, according to Avila-Stagno et al. [40].

### 2.8. Statistical Analysis

Data were analyzed using Stata version 16 statistical software (College Station, TX, USA). Shapiro–Wilk and Levene tests were used to verify the assumptions of normality (*p* > 0.05) and homogeneity of variances (*p* > 0.05), respectively. Radiata PBE concentration (0, 1, and 2% of dietary DM) was the main effect, and each individual lamb was considered as the experimental unit for hemogram values, serum biochemistry parameters, and carcass traits, whereas samples of each dietary treatment were the experimental unit for chemical composition of diets. The data from chemical composition of diets, hemogram values, serum biochemistry parameters, and carcass traits were subjected to analysis of variance (ANOVA) in a randomized complete block design, according to the following model:*Y_ij_* = *µ* + *α_i_* + *β_j_* + *ε_ij_,*

where *Y_ij_* is the observed value of the dependent variables; *µ* is the overall average; *α_i_* is the fixed treatment effect *i* (*i* = the effect of radiata PBE concentration); *β_j_* is the random effect of the repetition *j* (*j* = 3 for chemical composition of diets, and 9 for hemogram values, serum biochemistry parameters, and carcass traits); and *ε_ij_* is the experimental error.

The growth performance, intake, and nutrient conversion were analyzed as a randomized complete block design with repeated measures, according the following model:*Y*_*ijk*_ = *µ* + *α_i_* + *β_j_* + *δ_k_* + *ε_ijk_*

where *Y_ijk_* is the observed value of the dependent variables; *µ* is the overall average; α_i_ is the fixed effect of treatment *i* (*i* = the effect of radiata PBE concentration); *β_j_* is the random effect of the repetition *j* (*j* = 9); *δ_k_* is the random effect of time *k* (*k* = 21); and *ε_ijk_* is the experimental error. The results are presented as mean values with the standard error of the mean. The comparison of means was conducted by Tukey’s test, being statistically significant when *p* < 0.05. When fixed effect of treatment was significant, orthogonal polynomial contrasts were used to determine linear and quadratic responses to the radiata PBE concentration (0, 1, and 2% of dietary DM) and to compare control vs. radiata PBE supplementation for all variables [40]. None of the quadratic contrasts were significant, and are thus not reported.

## 3. Results

### 3.1. Chemical Composition of Diets

The radiata PBE inclusion decreased DM (*p* = 0.015) and CP (*p* = 0.013) linearly, and increased ADF (*p* = 0.007), TP (*p* < 0.001), and TT (*p* = 0.001) linearly. There were no differences (*p* ≥ 0.083) between dietary treatments in OM, EE, aNDF, HE, NFC, and FA profiles (Table 1).

### 3.2. Performance, Intake, and Nutrient Conversion

The RGR increased linearly (*p* = 0.008), with the radiata PBE supplementation (Table 2), being higher (*p* = 0.026) in 2% PBE. However, radiata PBE linearly improved (*p* = 0.003) the CPC. All other variables of the growth trial were unaffected (*p* ≥ 0.144) by radiata PBE inclusion in the diet.

### 3.3. Hematological and Serum Biochemistry Profile

Hemograms and serum biochemistry parameters were unaffected (Table 3) by radiata PBE supplementation (*p* ≥ 0.155 and *p* ≥ 0.106, respectively). According to the hemogram reference intervals (RI), only the mean corpuscular volume (MCV) was below the lower RI limit, whereas all lambs presented values below the lower RI limit in total bilirubin, globulins, and calcium (Ca), but above the upper limit in glucose, blood ureic nitrogen (BUN) and albumin in the biochemical profile.

### 3.4. Fatty Acid Profiles

The sum of SFAs (ΣSFA), proportions of myristic (14:0; MA), palmitic (16:0; PA), stearic (18:0; SA), elaidic (*t*9–18:1; EA) acids, Σn-6/Σn-3 ratio, and AI and TI were linearly reduced (*p* < 0.01) by radiata PBE supplementation (Table 4), whereas the ΣMUFA, ΣPUFA, ΣUFA, ΣPUFA/ΣSFA, Δ^9^C18, elongase, and the proportions of VA, oleic (*c*9–18:1; OA), linoleic (18:2 n-6; LA), alpha-linolenic (18:3 n-3; ALA), RA, CLA *t*10,*c*12–18:2, arachidic (20:0; ArA), arachidonic (20:4 n-6; AA), eicosapentaenoic (20:5 n-3; EPA), desirable FAs, and h:H ratio were linearly increased (*p* ≤ 0.016). The radiata PBE inclusion in the diets did not affect (*p* ≥ 0.112) Δ^9^C16 activity or the other FAs analyzed.

## 4. Discussion

We hypothesized that incorporating a polyphenol-rich extract from radiata pine bark in lamb finishing diets could improve animal performance and change FA profiles without detrimental effects on their DMI or health. This study demonstrated that a short-term (21-d) supplementation with radiata PBE improved sheep FA profile in adipose tissue (Figure 1) by (i) reducing ΣSFAs and the Σn-6/Σn-3 ratio, (ii) increasing the sum of unsaturated FAs (ΣUFA), and (iii) improving nutraceutical compounds; increasing CLA, h:H ratio, and desirable FAs and reducing EA, AI, and TI, which are beneficial to human health as AI and TI may reduce blood LDL levels [41]. Additionally, radiata PBE inclusion in the diet resulted in increased RGR and improved CPC, with unaffected weight gains, carcass traits, hemograms, and serum biochemistry.

### 4.1. Effects of Radiata PBE Supplementation on Fatty Acid Profile

The short-term supplementation with the radiata PBE increased ΣUFA proportions at the expense of ΣSFA, increasing the ΣPUFA/ΣSFA ratio and reducing the Σn-6/Σn-3 ratio, which suggests that radiata PBE could be used as an additive to improve the FA profile of ruminant-derived products given the increasing demand for quality milk and meat [42]. Ruminant-derived foods low in SFA and that contain PUFA and MUFA, in particular n-3 FA and CLA, are considered of general interest due to their beneficial effects on human health [17,43,44].

The increases in ΣPUFA, Σn-3, Σn-6, EPA, and AA acids found in this study agree with the meta-analysis of Torres et al. [4]; however, in contrast to those authors, this study revealed that radiata PBE supplementation increases RA. The ΣPUFA increase, especially alpha-linolenic and linoleic acids and EPA, suggest that radiata PBE polyphenols can inhibit lipolysis and/or the first steps of microbial BH by selective reduction or inhibition of rumen microorganisms [5]. One possible candidate to have been affected by radiata PBE is *Butyrivibrio fibrisolvens*, which metabolizes ALA, and LA to RA [43]. The increase in EPA proportions in subcutaneous fat of radiata PBE-supplemented lambs could be attributed to radiata PBE polyphenols deposited in fat tissue, which have the capability to enhance elongase enzyme activity [45]. The EPA increase in this study also concurs with the results obtained by Lee et al. [10] in goats supplemented with PB.

The increase in RA proportion could be due to direct increase in the dietary PUFA biohydrogenation, and/or could be synthesized in the animal’s tissues using VA as a precursor [17]. Rumenic acid is formed by BH of LA and is then converted to VA and subsequently to SA [42]. However, a greater proportion of RA is synthesized endogenously in tissues by VA desaturation via Δ^9^-desaturase [46]. Vasta et al. [47] reported that tannin supplementation increased the expression of Δ^9^-desaturase. Khiaosa-Ard et al. [17] and Vasta et al. [48] reported that CT supplementation inhibits *Butyrivibrio proteoclasticus* populations and the BH last step, leading to increases in VA at the expense of SA.

Reductions in ΣSFAs, SA and MA agree with the results of Fernandes et al. [44] in lambs fed with *Mimosa tenuiflora* as a source of tannins. The decrease in ΣSFAs in this study was mainly driven by the reduction in the proportions of SA and PA, which were found in greater proportions. Reductions in PA are considered beneficial for consumer health, as it is considered hypercholesterolemic [44]. The lower SFA ratio and the substitution of SA for OA in subcutaneous fat is associated with a higher rate of Δ^9^-desaturase activity [49] or to an inhibition of microorganisms involved in the BH of VA [46]. Increased proportions of LA and ΣPUFAs and the lower concentration of SA in the subcutaneous tissues are indicators of an inhibition of ruminal BH by the CT [47] present in the radiata PBE.

The ΣPUFA/ΣSFA ratio in this study was lower than 0.45, the recommended level for human diets [50]; however, radiata PBE increased this ratio from 0.1 to 0.2, which is comparable to the results of Costa et al. [51], who supplemented an *Acacia mearnsii* extract up to 80 g/kg in a concentrate diet for lambs. Radiata PBE increased Σn-3 and Σn-6 FA; however, Σn-6 FA was increased by 50% in diets supplemented with radiata PBE, whereas Σn-3 FA was increased by 80% and 100% with radiata PBE at 1 and 2% DM, respectively. This resulted in a linear reduction in the Σn-6/Σn-3 ratio, which is below the maximum recommendation of 5:1 [52], and is positive, as n-3 FA are essential for neurological development, have beneficial anti-inflammatory properties, and are associated with reduced risk of coronary heart disease [42].

The action of Δ^9^C18 was linearly increased by radiata PBE supplementation, which concurs with the results reported by Seoni et al. [53] in lambs fed CT-rich clover silage. This enzyme converts *c*11–18:1 and VA to RA in subcutaneous tissues of ruminants [54]. The increase in Δ^9^C18 could be partly attributed to an increased flux of 18:1 isomers into adipose tissue of lambs that consumed radiata PBE. The activity of elongase was also increased by radiata PBE supplementation, which coincides with the increase in AA and EPA, products of the action of Δ-desaturase-elongase on LA and ALA.

The AI was reduced substantially in radiata PBE supplemented lambs, results lower than the reference value of 1 for lambs [36], which concurs with the results of Pimentel et al. [55] in goats supplemented with acacia extract. An h:H ratio above 1.5% of FAME in mutton is considered healthy for humans [56]. Similarly, the linear decrease in TI would be considered healthy, since it results in a thrombogenic acid (myristic, palmitic, and stearic) decrease, and an antithrombogenic acid (Σn-6, Σn-3, and ΣMUFA) increase. In the present study, radiata PBE increased h:H ratio up to 1.75% when it was supplemented at 2% DM. The reference value of h:H ratio for meat products is 2.0 [37], which indicates products with a more desirable lipid profile, with lower cardiovascular risks. These indications, together with the changes obtained in the ratios of Σn-3/Σn-6 FA, ΣPUFA/ΣSFA, ΣPUFA, ΣMUFA, and the desirable FA indicator, indicate that supplementation with radiata PBE improves the lipid profile and nutritional quality of meat products derived from lamb. Future research is warranted to assess effects of longer supplementations in beef and dairy cattle and the combination with oilseeds.

### 4.2. Effects of Radiata PBE Supplementation on Performance and Efficiency

Unaffected DMI in radiata PBE-supplemented diets is in agreement with Norouzian and Ghiasi [57], who indicated that DMI did not decrease with the inclusion of polyphenol-rich pistachio by-products in growing lambs’ diets. Similarly, Krueger et al. [58] and Foiklang et al. [59] both reported that DMI did not decrease in animals supplemented with an acacia tannin extract (700 g CT/kg DM) or grape pomace, a high-tannin by-product, respectively. However, it is recognized that tannins reduce animals’ DMI by decreasing diet palatability due to short-term astringent taste, caused by the tannin’s bond to salivary proteins [58]. However, Dschaak et al. [60] reported a decrease in DMI and nutrients by cows fed diets containing quebracho CT extract, and Min et al. [6] found that the DMI of goats receiving PB (10.3% CT) was increased. The lack of consistency in the CT effects on DMI may be due to differences in the concentration, type, structure, and source of tannins, as well as by animal species and base diet [58]. As such, nutritional requirements of livestock seem to have a stronger impact on DMI as compared to CT [61].

The absence of radiata PBE effects on weight gain, final LW, and carcass traits suggests that the extract had no effect on lamb performance, which could be related to the lack of effect on DMI. Similar results on animal performance have been reported by the use of ground PB [9] and other sources of tannins such as acacia extract [58], pistachio by-products [57], and purple prairie clover hay [15]. Although there were no differences in ADG, radiata PBE supplementation increased the lambs’ RGR, indicating that supplementation could improve growth traits [26].

The dietary protein conversion may have been improved since there was no difference in the lambs’ final LW, and the CP intakes of radiata PBE-supplemented lambs were numerically lower than unsupplemented lambs. Thus, it can be inferred that the amount of available protein to be absorbed in the small intestine by lambs was increased as a result of radiata PBE supplementation. This is consistent with Min et al. [7], who reported that as ground PB (10.3% CT) increased in the diet, CP digestibility tended to decrease, but resulted in a positive impact on N-balance. Dschaak et al. [60] reported an improved feed efficiency in cows supplemented with a quebracho tannin extract (75% CT concentration) in high-forage diets, due to CT decreasing DM intake and rumen NH_3_-N concentration and not affecting milk protein yield. Similarly, Wang et al. [62] reported improved growth by improving rumen fermentation and N utilization in beef bulls by substitution of ground corn grain per steam-flaked sorghum grain (129.3 mg TT/g grain DM). In addition, Orlandi et al. [63] observed a decrease in urinary N excretion along with an improved amino acid supply of Holstein steers by the dietary inclusion of an acacia tannin extract (694 g TT/kg DM). However, Ahnert et al. [64] observed that increasing doses of a quebracho tannin extract (68.6% TT and 15.9% CT) in heifers’ diets linearly decreased N excretion in urine without improving animal performance. Discrepancy between results may be related to the effect of tannins on rumen microorganisms, resulting in decreased microbial protein synthesis, which would not be compensated by rumen escape protein [64]. Furthermore, tannin supplementation could lead to different amino acid profiles due to alterations on the final metabolizable protein pool, because dietary proteins that form complexes with CT would not be degraded compared to proteins without CT [16].

Considering that protein is the most expensive dietary component in most livestock systems, and that in many cases it is the limiting factor in animal production [16], the radiata PBE supplementation could have a positive impact in economic and environmental terms. The supplementation of ruminants with radiata PBE has the potential to reduce the cost of commercial ruminant diets by improving the CPC. Future research is warranted to assess radiata PBE supplementation in diets with lower protein content than the ones used in this study and to evaluate the economic efficiency and urine N excretion in supplemented animals.

### 4.3. Effects of Radiata PBE Supplementation on Hemogram and Serum Biochemistry

Hematological and serum biochemistry analysis are an important support to evaluate the animals nutritional and health status [29]. These results confirm the lambs’ health status in this study. Both the hematological and serum biochemistry values were unaffected by supplementation, suggesting that concentrations of 1 and 2% DM of radiata PBE do not produce hepatic or renal damage or have a negative impact on the immune response or health of Suffolk lambs.

Laboratory analysis are not a substitute for clinical examination and management practices analysis [28,29]. During slaughter of the animals, liver and kidneys were inspected (data not submitted), corroborating laboratory data. Similar results have been obtained with other polyphenol sources in lambs [15] and goats [6,9]. In contrast, Wang et al. [62] found an improvement in blood metabolites by the inclusion of steam-flaked sorghum grain, with moderate levels of tannins, in young cattle diets. However, Acharya et al. [65] reported subtle but significant differences in the hematological and serum biochemistry analysis of lambs fed Sericea lespedeza (6–10% CT) compared with control lamb, while in dairy steers, Foiklang et al. [59] observed an increase in BUN concentration by grape pomace.

Compared to RI values, animals had a slightly low MCV. The most likely cause may be the age of the lambs (4 months old), in which the MCV value may be lower than in adult animals [66]. The increased glucose in all lambs may be related to stress during the blood collection process, since in stressful situations adrenaline may increase the rate of muscle glycogen turnover, causing muscle glycogen mobilization [67]. Increases in blood albumin and BUN concentration are likely associated with protein contents of the diets that were formulated to simulate high-protein intakes in temperate spring pastures in Chile. Intake of a CP-rich diet increases the BUN concentration due to the higher urea metabolization in the liver from rumen ammonia (NH_3_) absorption. Blood albumin concentration may increase because the liver uses dietary protein to synthesize albumin [65]. It was expected that total bilirubin would be low, as even in severe liver damage, the magnitude of the ruminants’ total bilirubin concentration increase is not very high [29]. The slight Ca reduction may be related to the ingredients of the base diet, as some factors that decrease blood Ca concentration are the use of grains of wheat in feed [68]. Globulin concentration was also slightly below the RI, which could be related to a developing immune system as the hemogram and total protein concentration were within the RI [69], however, it is possible that the total protein concentration was normal due to the increase in albumin.

### 4.4. Effects of Radiata PBE Supplementation on the Chemical Composition of Diets

The supplementation with radiata PBE changed the diets’ composition, decreasing DM and CP and increasing ADF. The decrease in DM is due to radiata PBE being aqueous (38.0% DM), and this effect has been previously reported [12,13,14]. Likewise, the decrease in CP could be related to the decrease in DM. Vera et al. [13] showed a numerical decrease in DM and CP of high-concentrate diets, supplemented with concentrations < 2% DM of PBE. Regarding the ADF increase, it may be associated to the amount of ADF and lignin (64.9 and 45.1% DM, respectively) of the radiata PBE, similar to that reported by other studies with pine bark [6,7] and PBE [12].

The present study highlighted that a short-term supplementation with PBE improves subcutaneous FA profiles, and has the potential to improve RGR and CPC, with unaffected intake, growth performance, blood parameters, or carcass characteristics. However, further studies are needed, using younger animals to confirm the PBE effect on RGR and CPC and to clarify the effects of long-term supplementation on FA profile for intramuscular, intermuscular, and subcutaneous fat and lipid oxidation of meat, in addition to exploring the PBE effects on the environmental impact of ruminants.

## 5. Conclusions

Results indicate that a short-term supplementation with 1 or 2% DM of a polyphenol-rich extract from radiata pine bark (PBE) can modulate ruminal biohydrogenation and improve subcutaneous fatty acid profiles by increasing conjugated linoleic acid (*c*9,*t*11–18:2 and *t*10,*c*12–18:2), vaccenic (*t*11–18:1) and oleic (*c*9–18:1) acids, and total n-3 fatty acids, and reducing total saturated fatty acids, elaidic acid (*t*9–18:1), total n-6/total n-3 ratio, atherogenicity index, and thrombogenic index, without affecting dry matter intake, growth performance, blood parameters, or carcass traits. Additionally, radiata PBE supplementation enhances relative growth rate and dietary protein conversion, which suggests that radiata PBE has the potential to be used as an environmentally friendly additive that reduces the cost of commercial ruminant diets.

## Figures and Tables

**Figure 1 animals-13-00188-f001:**
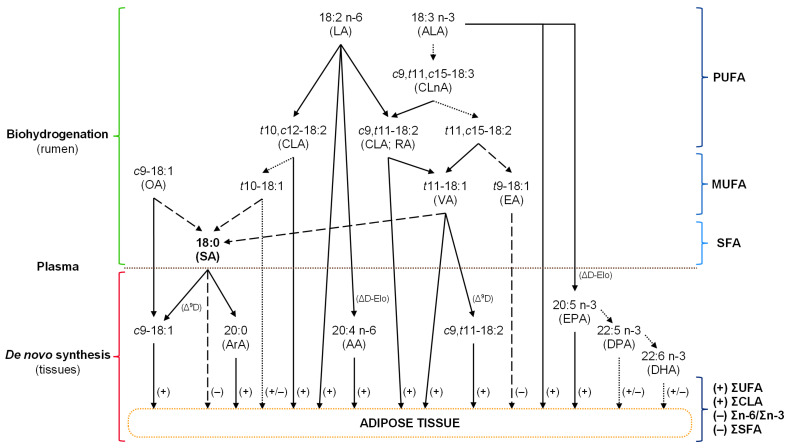
Potential pathways of biohydrogenation and biosynthesis of dietary fatty acids under the supplementation with a polyphenol-rich extract from radiata pine bark. Arrows with solid, dashed, and dotted lines highlight the possible increase, decrease, or maintenance of the biohydrogenation and biosynthesis products, respectively. Signs “+” and “–” indicate increase and decrease in fatty acids percentage in adipose tissue, respectively. Δ9D = delta-9-desaturase; ΔD-Elo = desaturase-elongase; AA = arachidonic acid; ALA = alpha-linolenic acid; ArA = arachidic acid; CLA = conjugated linoleic acid; CLnA = conjugated linolenic acid; DHA = docosahexaenoic acid; DPA = docosapentaenoic acid; EA = elaidic acid; EPA = eicosapentaenoic acid; LA = linoleic acid; MUFA = monounsaturated fatty acids; OA = oleic acid; PUFA = polyunsaturated fatty acids; RA = rumenic acid; SA = stearic acid; SFA = saturated fatty acids; UFA = unsaturated fatty acids; VA = vaccenic acid.

**Table 1 animals-13-00188-t001:** Ingredients, chemical composition, and fatty acid profile of dietary treatments fed to Suffolk lambs.

Item	Radiata PBE, % DM			SEM	*p-*Value ^1^		
	0	1	2		Diet	Linear ^2^	0 vs. PBE
Ingredients, g/kg DM							
Alfalfa hay	437.0	432.6	428.3	–	–	–	–
Soybean meal	146.0	144.5	143.1	–	–	–	–
Wheat bran	175.0	173.3	171.5	–	–	–	–
Corn grain	233.0	230.7	228.3	–	–	–	–
Mineral premix ^3^	9.0	8.9	8.8	–	–	–	–
Pine bark extract	–	10.0	20.0	–	–	–	–
Chemical composition, % DM (*n* = 3)							
Dry matter, % FW	89.2 ^a^	87.5 ^ab^	87.1 ^b^	0.48	0.032	0.015	0.001
Organic matter	92.6	93.0	93.1	0.18	0.083	–	–
Crude protein	20.2 ^a^	18.5 ^b^	18.7 ^b^	0.34	0.014	0.013	0.003
Ether extract	2.29	2.26	2.22	0.058	0.741	–	–
Neutral detergent fiber	33.4	35.1	35.2	0.90	0.247	–	–
Acid detergent fiber	20.9 ^b^	23.7 ^ab^	24.4 ^a^	0.77	0.017	0.007	0.012
Hemicellulose	12.5	11.5	10.3	1.37	0.566	–	–
Nonfibrous carbohydrates ^4^	36.7	37.2	37.4	0.99	0.881	–	–
Phenolic constituents, % DM							
Total polyphenols	0.64 ^b^	1.35 ^b^	2.61 ^a^	0.197	<0.001	<0.001	0.014
Total tannins	0.19 ^b^	0.26 ^ab^	0.34 ^a^	0.027	0.004	0.001	0.016
Fatty acids, g/100 g fatty acids							
16:0	20.2	19.9	19.7	0.03	0.083	–	–
18:0	3.59	3.55	3.52	0.005	0.102	–	–
*c*9–18:1	14.5	14.3	14.2	0.05	0.149	–	–
*c*11–18:1	0.50	0.49	0.49	0.002	0.274	–	–
18:2 n-6	20.8	20.5	20.3	0.05	0.156	–	–
18:3 n-3	16.7	16.5	16.3	0.04	0.140	–	–

DM = dry matter; FW = fresh weight; PBE = pine bark extract; SEM = standard error of the mean; ^1^ when diet effects are not significant (*p* > 0.05), values for contrasts are not reported; ^2^ linear effects of increasing concentrations of radiata PBE; ^3^ mineral premix provided macrominerals (g/kg of Ca 37.8; P 34.5; S 6.0; K 4.7; Mg 2.4 and 1.0 Fe), and microminerals (mg/kg of Zn 750; Cu 40; I 20; Co 4 and Mn 1); ^4^ nonfibrous carbohydrates = organic matter—crude protein—ether extract—neutral detergent fiber [19]; ^a,b^ Within a row, means with different superscripts differ significantly (*p* < 0.05), as determined by Tukey’s test.

**Table 2 animals-13-00188-t002:** Effects of polyphenol-rich extract from radiata pine bark concentration (0, 1, and 2% of dietary DM) on performance, efficiency, and carcass characteristics of Suffolk lambs (*n* = 9).

Item	Radiata PBE, % DM			SEM	*p-*Value ^1^		
	0	1	2		Diet	Linear ^2^	0 vs. PBE
Performance							
Initial live weight, kg	35.7	36.1	36.3	1.22	0.923	–	–
Final live weight, kg	42.2	43.3	42.7	1.57	0.873	–	–
Average daily gain ^3^, g/d	292	325	334	39.8	0.679	–	–
Dry matter intake, g/d	1718	1732	1796	45.9	0.402	–	–
Crude protein intake, g/d	340	320	321	8.7	0.144	–	–
Efficiency							
Feed conversion ratio ^4^, g/g	5.39	4.74	4.65	0.587	0.615	–	–
Crude protein conversion ^5^, g/g	1.35 ^a^	0.89 ^b^	0.85 ^b^	0.124	0.006	0.003	0.002
Relative growth rate ^6^	0.26 ^b^	0.36 ^ab^	0.41 ^a^	0.045	0.026	0.008	0.007
Carcass characteristics							
Hot carcass weight, kg	21.7	22.5	21.3	0.83	0.546	–	–
Hot carcass yield ^7^, %	50.3	51.2	50.0	0.70	0.116	–	–
Cold carcass weight, kg	21.1	22.0	20.7	0.79	0.537	–	–
Cold carcass yield ^8^, %	49.5	50.0	48.3	0.58	0.118	–	–
Weight losses by cooling ^9^, %	2.67	2.73	2.75	0.108	0.864	–	–

DM = dry matter; PBE = pine bark extract; SEM = standard error of the mean; ^1^ when diet effects are not significant (*p* > 0.05), values for contrasts are not reported; ^2^ linear effects of increasing concentrations of radiata PBE; ^3^ average daily gain = (final live weight–initial live weight)/trial duration in days [26]; ^4^ feed conversion ratio = dry matter intake/average daily gain [26]; ^5^ crude protein conversion = crude protein intake/average daily gain [26]; ^6^ relative growth rate = [(log final live weight—log initial live weight)/trial duration in days] × 100 [27]; ^7^ hot carcass yield = (hot carcass weight/final live weight) × 100 [32]; ^8^ cold carcass yield = (cold carcass weight/final live weight) × 100 [32]; ^9^ weight losses by cooling = [(hot carcass weight—cold carcass weight)/hot carcass weight] × 100 [32]; ^a,b^ within a row, means with different superscripts differ significantly (*p* < 0.05), as determined by Tukey’s test.

**Table 3 animals-13-00188-t003:** Effects of polyphenol-rich extract from radiata pine bark concentration (0, 1, and 2% of dietary DM) on hemogram values and serum biochemistry parameters of Suffolk lambs (*n* = 9).

Item	RI ^1^	Radiata PBE, % DM			SEM	*p*-Value ^2^
		0	1	2		Diet
		Hemogram				
Hematology						
Red blood cells, 10^6^/μL	9–15	12.5	12.2	12.5	0.42	0.720
Hemoglobin, g/dL	9–15	11.2	11.2	11.5	0.30	0.701
Hematocrit, %	27–45	32.6	32.4	33.5	0.85	0.503
MCV, fL	28–40	26.1	26.7	26.7	0.53	0.634
MCH, g/dL	8–12	9.0	9.2	9.1	0.17	0.703
MCHC, %	31–34	34.4	34.5	34.1	0.36	0.713
Platelets, 10^3^	100–800	530	584	557	86.7	0.891
White blood cells, 10^3^/μL	4–12	10.7	11.3	10.1	1.62	0.789
Differential, %						
Segmented neutrophils	10–50	24.1	27.4	21.2	3.57	0.473
Lymphocytes	40–75	67.4	74.3	68.3	3.58	0.341
Eosinophils	0–10	2.0	0.7	1.6	0.50	0.155
Monocytes	0–6	2.6	2.6	2.8	0.33	0.834
Basophils	0–3	1.5	0.3	0.0	0.65	0.278
		Serum biochemistry				
Metabolites						
Total bilirubin, mg/dL	0.1–0.5	0.03	0.04	0.03	0.005	0.229
Glucose, mg/dL	50–80	88.7	88.6	88.0	2.71	0.980
Total proteins, g/dL	6.0–7.9	6.7	6.5	6.4	0.12	0.412
Albumin, g/dL	2.4–3.0	3.7	3.8	3.7	0.04	0.366
Globulins, g/dL	3.5–5.7	3.0	2.7	2.7	0.13	0.378
Albumin:globulin ratio	–	1.27	1.44	1.38	0.068	0.217
Creatinine, mg/dL	1.2–1.9	1.02	0.96	1.02	0.038	0.382
BUN, mg/dL	8–20	31.6	28.8	31.1	1.65	0.448
Enzymes, IU/L						
Alkaline phosphatase	68–387	153	154	153	11.8	0.998
Aspartate aminotransferase	60–280	123	113	112	6.7	0.440
Alanine aminotransferase	≤188	20.3	17.3	18.0	1.03	0.106
Minerals, mEq/L						
Calcium	11.5–12.8	9.7	10.1	9.9	0.12	0.160
Phosphorus	5.0–7.3	7.0	6.1	6.3	0.42	0.350

BUN = blood ureic nitrogen; DM = dry matter; MCH = mean corpuscular hemoglobin; MCHC = mean corpuscular hemoglobin concentration; MCV = mean corpuscular volume; PBE = pine bark extract; SEM = standard error of the mean; ^1^ reference interval [29]; ^2^ when diet effects are not significant (*p* > 0.05), values for contrasts are not reported.

**Table 4 animals-13-00188-t004:** Effects of polyphenol-rich extract from radiata pine bark concentration (0, 1, and 2% of dietary DM) on fatty acid profile in adipose tissue of Suffolk lambs (*n* = 9).

Item	Radiata PBE, % DM			SEM	*p-*Value ^1^		
	0	1	2		Diet	Linear ^2^	0 vs. PBE
Saturated fatty acids							
10:0	0.55	0.56	0.55	0.021	0.764	–	–
12:0	0.35	0.34	0.31	0.035	0.642	–	–
14:0 (MA)	3.80 ^a^	2.70 ^b^	2.51 ^b^	0.287	0.011	0.005	0.003
15:0	1.35	1.34	1.41	0.132	0.898	–	–
16:0 (PA)	26.1 ^a^	25.1 ^a^	22.2 ^b^	0.53	<0.001	<0.001	0.013
17:0	2.14	1.90	2.14	0.101	0.206	–	–
18:0 (SA)	18.7 ^a^	14.9 ^b^	14.6 ^b^	0.79	0.001	0.001	<0.001
20:0 (ArA)	1.30 ^b^	1.42 ^ab^	1.51 ^a^	0.051	0.031	0.010	0.013
∑SFA	53.6 ^a^	47.8 ^b^	46.4 ^b^	1.07	<0.001	<0.001	<0.001
Monounsaturated fatty acids							
16:1	1.97	1.84	1.83	0.122	0.596	–	–
*t*6,*t*8–18:1	0.28	0.21	0.28	0.042	0.354	–	–
*t*9–18:1 (EA)	0.43 ^a^	0.34 ^b^	0.34 ^b^	0.018	0.032	0.018	0.004
*t*10–18:1	1.25	1.29	1.37	0.080	0.287	–	–
*t*11–18:1 (VA)	2.16 ^b^	2.46 ^ab^	2.52 ^a^	0.103	0.025	0.010	0.002
*c*9–18:1 (OA)	31.4 ^b^	34.9 ^a^	35.7 ^a^	0.98	0.006	0.002	0.001
*c*11–18:1	1.53	1.53	1.54	0.036	0.991	–	–
∑MUFA	39.3 ^b^	42.8 ^ab^	44.0 ^a^	1.14	0.010	0.003	0.001
Polyunsaturated fatty acids							
18:2 n-6 (LA)	2.94 ^b^	4.41 ^a^	4.52 ^a^	0.299	0.002	0.001	<0.001
18:3 n-3 (ALA)	0.25 ^b^	0.28 ^ab^	0.41 ^a^	0.041	0.029	0.013	0.045
CLA *t*10,*c*12–18:2	0.58 ^b^	0.72 ^ab^	0.90 ^a^	0.098	0.046	0.015	0.037
CLA *c*9,*t*11–18:2 (RA)	0.64 ^b^	0.79 ^ab^	0.88 ^a^	0.072	0.041	0.016	0.008
20:4 n-6 (AA)	0.78 ^b^	0.92 ^ab^	1.10 ^a^	0.098	0.046	0.015	0.037
20:5 n-3 (EPA)	0.05 ^b^	1.02 ^a^	1.11 ^a^	0.193	0.012	0.006	0.046
22:5 n-3 (DPA)	0.25	0.31	0.35	0.060	0.353	–	–
22:6 n-3 (DHA)	0.09	0.08	0.10	0.054	0.935	–	–
∑n-3	1.04 ^b^	1.88 ^ab^	2.11 ^a^	0.271	0.034	0.009	0.001
∑n-6	3.43 ^b^	5.37 ^a^	5.62 ^a^	0.320	0.002	<0.001	<0.001
TFA—(CLA + VA) ^3^	1.25	1.10	1.12	0.119	0.247	–	–
∑PUFA	5.74 ^b^	8.48 ^a^	9.30 ^a^	0.613	0.001	<0.001	<0.001
∑UFA ^4^	44.6 ^b^	51.0 ^a^	53.0 ^a^	1.61	0.001	<0.001	<0.001
Ratio							
∑n-6/∑n-3 ^5^	3.84 ^a^	2.75 ^ab^	2.37 ^b^	0.264	0.020	<0.001	0.008
∑PUFA/∑SFA	0.10 ^b^	0.17 ^a^	0.20 ^a^	0.146	0.002	<0.001	<0.001
Nutraceutical compounds							
Desirable fatty acids ^6^	61.2 ^c^	65.1 ^b^	69.2 ª	0.92	0.001	0.001	0.001
Atherogenicity index ^7^	0.92 ^a^	0.72 ^b^	0.64 ^b^	0.469	< 0.001	<0.001	<0.001
Thrombogenic index ^8^	1.76 ^a^	1.27 ^b^	1.12 ^b^	0.055	< 0.001	<0.001	<0.001
h:H ratio ^9^	1.23 ^c^	1.52 ^b^	1.75 ^a^	0.670	< 0.001	<0.001	<0.001
Delta-9-desaturase C16 ^10^	6.90	6.69	7.75	0.406	0.112	–	–
Delta-9-desaturase C18 ^11^	62.4 ^b^	70.9 ^a^	71.6 ^a^	1.393	0.001	0.001	<0.001
Elongase ^12^	65.7 ^b^	66.4 ^b^	69.5 ^a^	0.668	< 0.001	0.015	0.023

*c* = cis; CLA = conjugated linoleic acid; DM = dry matter; FA = fatty acids; h:H ratio = hypocholesterolemic and hypercholesterolemic fatty acid ratio; MUFA = monounsaturated fatty acids; PBE = pine bark extract; PUFA = polyunsaturated fatty acids; SEM = standard error of the mean; SFA = saturated fatty acids; *t* = trans; UFA = unsaturated fatty acids; TFA = trans fatty acids; ^1^ when diet effects were not significant (*p* > 0.05), values for contrasts were not reported; ^2^ linear effects of increasing concentrations of radiata PBE; ^3^ TFA – (CLA +VA) = (*t*6,*t*8–18:1 + *t*9–18:1 + *t*10–18:1 + *t*11–18:1) – (*c*9,*t*11–18:2 + *t*10,*c*12–18:2 + *t*11–18:1) [40]; ^4^ ∑UFA = ΣMUFA + ΣPUFA [40]; ^5^ ∑n-6/∑n-3 = (18:2 n-6 + 20:4 n-6)/(18:3 n-3 + 20:5 n-3 + 22:5 n-3 + 22:6 n-3) [40]; ^6^ desirable fatty acids = (ΣMUFA + ΣPUFA + 18:0) [38]; ^7^ atherogenicity index = [(12:0 + (4 × 14:0) + 16:0)]/(ΣMUFA + Σn-6 + Σn-3) [36]; ^8^ thrombogenic index = (14:0 + 16:0 + 18:0)/[0.5 × ΣMUFA + 0.5 × Σn-6 + 3 × Σn-3)) + ∑n-3/∑n-6] [36]; ^9^ h:H ratio = (*c*9–18:1 + 18:2 n-6 + 20:4 n-6 + 18:3 n-3 + 20:5 n-3 + 22:5 n-3 + 22:6 n-3)/(14:0 + 16:0) [37]; ^10^ delta-9-desaturase C16 = [16:1/(16:0 + 16:1)] × 100 [39]; ^11^ delta-9-desaturase C18 = [(*c*9–18:1)/(18:0 + *c*9–18:1)] × 100 [39]; ^12^ elongase = [(18:0 + *c*9–18:1)/(16:0 + 16:1 + 18:0 + *c*9–18:1)] × 100 [39]; ^a,b,c^ within a row, means with different superscripts differ significantly (*p* < 0.05), as determined by Tukey’s test.

## Data Availability

The data for this study are available on request from the corresponding author.

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
