# Peer review of "A Short-Term Supplementation with a Polyphenol-Rich Extract from Radiata Pine Bark Improves Fatty Acid Profiles in Finishing Lambs"

_animals, 2023, doi:10.3390/ani13020188_

Round 1
Reviewer 1 Report
The study is well designed and executed, and provides additional insights on the effect of CT on the nutritional impact and performance of lambs. The manuscript is well written and the discussion places the findings of this study in the appropriate relation to the existing literature. Two minor points should be addressed by the authors:
The description of the lambs used for the study is confusing (17 lambs and 10 ewes). ‘Ewes’ would be considered mature/older animals; what was the range in ages around the average age of 4 months? If the differentiation between lambs and ewes refers to sex, these should be listed as male and female, ram (wether) and ewe lambs. Using differing sex classes for the study would likely have had an effect on performance and/or fatty acid profiles, unless groups were balanced accordingly.
The authors introduce ‘relative growth rate’ as a measure of performance. Was this done to account for the relatively short feeding duration? If though this should be stated. Not sure how much value should be placed on this measure with none of the other animal performance measures showing any treatment differences. Should the reference or relative growth rate be associated with Fitzhugh and Taylor, 1971, rather than Berry and Crowley, 2013?
Reviewer 2 Report
Very nice design and clear, concise presentation of an extremely timely topic, congratulations on execution and interpretations.
Noted only 2 extremely minor issues in the writing: 1) a word is missing in line 365 should read "may be due to" instead of "may due to" and 2) in line 429 should the word be "temperate" instead of "template"?
Reviewer 3 Report
Dear Authors,
the submitted manuscript needs a thorough review before being published. Here are some observations that may help you in revising the manuscript.
Introduction
The paragraph needs to be implemented. Even if the studies relating to the use of PBEs are limited, reference must also be made to the use of other matrices, highlighting the effect of polyphenols and condensed tannins on livestock production.
Materials and methods
Line 96, you state that you use 27 lambs (17 lambs and 10 ewes),. What do you mean? 17 males and 10 females? explain the concept better.
Line 101, you abstractly state that PBE is added as an extract from... (lines 29-30). In this part you should specify how to obtain the extract as well as how to integrate it into the control diet (liquid extract, dehydrated,...)
Table 1, lacking information on how PBE was integrated into the control diet, I wonder how this could have changed the content in DM, CP and ADF. Furthermore, if the integration of 1% of PBE determined an increase of total polyphenols equal to 0.71% DM (1.35-0.64), how does the integration of 2% of PBE determine an increase of 1.97% DM (2.61-0.64 ), I would expect a lower value. Finally, I believe it is more appropriate to report the averages with the relative standard deviations of the results of the chemical analyzes of the three diets.
Results
They need to be improved and rearranged to make them clearer. Much of the attention is focused on highlighting the effect on the linearity of the treatments on the variables. Finally, it would be appropriate to also report the values of the averages.
Line 207, report that the treatments resulted in a linear reduction on DM and CP but report only the P-value of DM. Same thing on lines 208, 209 and 213.
Discussions
You must respect the same sequence with which you approach the paragraph of the Results.
Lines 266-275, is this an abstract of the discussions paragraph? I think it should be deleted
Conclusions
They need to be expanded and improved.
Best regards
Reviewer 4 Report
The authors studied the effects of a short-term supplementation with a polyphenol-rich extract from radiata pine bark (PBE) on animal performance, blood parameters, and fatty acid (FA) profiles in finishing lambs. This manuscript is interesting. However, there are some issues that need to be addressed. My comments are below.
1. Line 19: “a polyphenolic-rich extract from radiata pine bark (PBE) has demonstrated to...”, letter “a” should be capitalized.
2. Is the scope of "protein conversion" in the title too wide?
3. Line 29-31: The title revealed “protein conversion”, but the purpose of the abstract is not clear.
4. Line 45: I suggest deleting the "by-product "in keywords.
5. Line 96-98: I'm curious about the gender, how many rams and ewes in each group, and whether gender is taken into account?
6. Line 103: The word “meet” should be revised to “met”.
7. Is there fasting before slaughter, please supplement.
8. Why do you choose to feed 35 days for slaughter? How do you define short-term?
9. Have you mentioned how to harvest adipose tissue in “Materials and Methods” ? Please supplement details. When was the adipose tissue collected after animal slaughter and was it oxidized? I think this is very important. And what part of the adipose tissue was selected, subcutaneous or perirenal?
10. Line 273-274: “Additionally, radiata PBE inclusion in the diet resulted in increased in RGR and improved CPC”, please check if this sentence is correct.
11. Line 310-311: “Vasta et al. [46] reported that tannins supplementation increases the expression of Δ9-desaturase”. The “increases” should be revised to “increased”, is the change right?
12. Please check the tense of the whole text.
13. Line 368-370: “The initial LW of lambs was similar among treatments, thus, the absence of ...”. I don't think this sentence is correct. Is the “thus” used appropriate?
14. Please briefly state what you suggest should be the next step based on the existing experimental results.
Round 2
Reviewer 3 Report
Dear Authors,
thank you for editing your manuscript, in this way the understanding of the materials and methods will be clearer.
With regard to the comments on the discussion, I remain of my opinion. I agree that the reader does not read the entire article in order, but precisely for this reason if I read a part of the results I don't have to read all the discussions to find the part relating to those results.
I have not attended courses on writing high-impact scientific articles, but I have certainly read and published several articles in high-impact journals. Furthermore, a manuscript must be read entirely to have a complete view of the experiment and its implications.
Finally, do not report the salient values in the results to avoid weighing down the reading and being redundant but repeat an entire paragraph as it allows the reader to know, or remember, our research hypothesis and its most important results, it seems to me in contrast.
Best regards
Reviewer 4 Report
Nothing